# Synthesis and Characterizations of Zinc Oxide on Reduced Graphene Oxide for High Performance Electrocatalytic Reduction of Oxygen

**DOI:** 10.3390/molecules23123227

**Published:** 2018-12-06

**Authors:** Jiemei Yu, Taizhong Huang, Zhankun Jiang, Min Sun, Chengchun Tang

**Affiliations:** 1School of Materials Science and Engineering, Hebei University of Technology, 8 of First Road of Dingzigu, Hongqiao District, Tianjin 300130, China; chm_yujm@ujn.edu.cn; 2School of Chemistry and Chemical Engineering, University of Jinan, 336 West Nanxinzhuang Road, Jinan 250022, China; chm_huangtz@ujn.edu.cn (T.H.); chm_jiangzk@ujn.edu.cn (Z.J.)

**Keywords:** non-precious-metal catalyst, oxygen reduction reaction, zinc oxide, reduced graphene oxide

## Abstract

Electrocatalysts for the oxygen reduction (ORR) reaction play an important role in renewable energy technologies, including fuel cells and metal-air batteries. However, development of cost effective catalyst with high activity remains a great challenge. In this feature article, a hybrid material combining ZnO nanoparticles (NPs) with reduced graphene oxide (rGO) is applied as an efficient oxygen reduction electrocatalyst. It is fabricated through a facile one-step hydrothermal method, in which the formation of ZnO NPs and the reduction of graphene oxide are accomplished simultaneously. Transmission electron microscopy and scanning electron microscopy profiles reveal the uniform distribution of ZnO NPs on rGO sheets. Cyclic voltammograms, rotating disk electrode and rotating ring disk electrode measurements demonstrate that the hierarchical ZnO/rGO hybrid nanomaterial exhibits excellent electrocatalytic activity for ORR in alkaline medium, due to the high cathodic current density (9.21 × 10^−5^ mA/cm^2^), positive onset potential (−0.22 V), low H_2_O_2_ yield (less than 3%), and high electron transfer numbers (4e from O_2_ to H_2_O). The proposed catalyst is also compared with commercial Pt/C catalyst, comparable catalytic performance and better stability are obtained. It is expected that the ZnO/rGO hybrid could be used as promising non-precious metal cathode in alkaline fuel cells.

## 1. Introduction

The oxygen reduction reaction (ORR) at the cathode plays crucial rule in energy transformation, for it enables various energy technologies to achieve large energy densities, including water splitting, metal-air batteries and fuel cells [1,2]. Over the past decades, Pt-based catalysts have shown the best catalytic performance for ORR. Nevertheless, such materials suffer from poor reliability and durability problems (e.g., poisoning and crossover effects), as well as the low abundance and the increasing cost of platinum, which has constrained their large-scale commercial application [3,4,5]. Thus, searching for cheap and abundant catalysts with excellent performance substituting for Pt is very important and necessary from the point view of sustainable development [6,7]. 

Graphene, one of conductive carbon matrices, has attracted continually attention because of its high surface area, superior electric conductivity, large surface area, good flexibility, and strong adhesion to catalytic particles [8,9,10]. With the unique surface chemistry and 2D microstructure, graphene has been proven to be an ideal platform for anchoring or growing guest species [11,12]. Thus, many new strategies for growing metal oxide layers on graphene nanosheets have been developed. Recent investigations have demonstrated that transition metal oxide nanoparticles (NPs) anchored on graphene are one of the most promising ORR catalysts, which can substitute for Pt and Pt-based catalysts. Nitrogen-doped reduced graphene oxide supported cobalt oxide nanosheets [13], Co@CoO and acetylene black particles supported on the wrinkled nitrogen doped reduced graphene oxide (rGO) [14], Co_3_O_4_ nanosheets anchored on rGO-coated nickel foam [15], spinel CoMn_2_O_4_ doped reduced graphene oxide [16], (*N*,*F*)-codoped TiO_2_ supported by rGO [17], and MnO_2_ nanoparticles anchored on rGO [18] et al., have exhibited comparable electrocatalytic activity but superior stability toward ORR compared with the commercial Pt/C catalyst. These studies on transition metal oxides-graphene interactions revealed that depending on the metal-graphene spacing and the spinel structure of transition metal oxides, charge transfer may exist between the transition metal oxides and graphene, which may be the real reason that certain transition metal oxides NPs supported on graphene surface exhibit enhanced catalytic activities [19].

With the high desirability for seeking clean and efficient energy sources, there is still room for developing cost effective ORR catalysts with high catalytic properties. Among the transition metal oxides that have been investigated, Zinc oxide (ZnO), with its outstanding merits of high catalytic activity, non-toxicity and great chemical stability, has drawn particular attention [20,21]. Over the past decades, ZnO nanostructures have been explored for applications such as field effect transistors, ultra-violet light sources, piezoelectric nanogenerators, gas sensors, cold cathode electron sources and solar cells [22,23,24]. However, up to now there has been no report about ZnO serving as an efficient ORR catalyst. Considering the relatively poor electroconductivity of ZnO, which may greatly hinder its application as an electrode material in the oxygen reduction process, it is necessary to combine ZnO with rGO layers as oxygen reduction catalyst. Therefore, the electrocatalyst of ZnO NPs anchored on rGO was obtained through an extremely facile route in this paper. We hope it can be used as a highly efficient electrocatalyst.

## 2. Results and Discussion

### 2.1. Structural Characterization

Figure 1a shows the XRD (X-ray diffraction) patterns of ZnO and ZnO/rGO, the characteristic peak locates at 31.78°, 34.43°, 36.26°, 47.55°, 56.61°and 62.88°, corresponding to the (100), (002), (101), (102), (110) and (103) plane of ZnO with the hexagonal wurtzite phase (PDF: 01-089-0510), respectively [25]. No evidence of any impurities were detected. The XRD pattern of ZnO/rGO also shows an unconspicuous peak at around 24.2°, which related to the (002) plane of graphene, implying the removal of oxygenated functional groups during the reduction of GO into rGO. Figure 1b shows the Raman spectrum of ZnO/rGO. The D band (1350 cm^−1^) is correlated to the disorders and structural defects of graphene, whereas the G band (1590 cm^−1^) is corresponded to the sp^2^-hybridized graphitic carbon atoms [26,27]. The relative intensity rations of D to G peak (I*_D_*/I*_G_*) for ZnO/rGO is 1.1, indicating the partially reduced of GO to rGO.

X-ray photoelectron spectroscopy (XPS) analyses were recorded to probe the elemental chemical states and the surface chemical composition of our samples. Figure 1c displays the survey spectra of ZnO and ZnO/rGO. As expected, Figure 1c clearly shows that the ZnO/rGO mainly contains the elements C, O and Zn. The C element of ZnO/rGO results from the adventitious element carbon and the rGO molecules, while in the ZnO spectrum, it only comes from the contamination of the testing environment. Figure 1d shows the C 1s spectrum of ZnO/rGO hybrid, which can be deconvoluted into three components identified as C-C (284.0 eV), C-O (286.0 eV) and C=O (287.5 eV), respectively. The Zn 2p spectrum of the ZnO (Figure 1e) and ZnO/rGO (Figure 1f) displays doublet peaks. The peak located at 1044.1 eV and 1021.5 eV are assigned to Zn 2p_1/2_ and 2p_3/2_ of Zn^2+^ in ZnO, while the corresponding peaks of Zn 2p in ZnO/rGO were observed at 1045.0 eV and 1022.4 eV, shifting towards the higher energy region by 0.9 eV [28]. The O 1s XPS spectrum of ZnO and ZnO/rGO is shown in Figure 1g,h. As seen in Figure 1g, the peaks located at 530.1 eV and 532.3 eV correspond to O^2−^ ions in the Zn-O species and the oxygen-deficient regions of the wurtzite ZnO structure, respectively [29]. The peak located at 534.1 eV is related to the OHs or oxygen groups absorbed onto the surface of the catalyst [30]. The corresponding O 1s peaks in ZnO/rGO are located at 530.3 eV, 532.5 eV and 534.2 eV. It should be mentioned that the peak intensities at 532.5 eV and 534.2 eV of ZnO/rGO are much stronger than that of pure ZnO, indicating the significant increase of OH or oxygen groups absorbed onto the surface of ZnO/rGO, compared with pure ZnO. The absorbed oxygen or OH groups may be helpful to enhance the electrocatalytic activity of ZnO/rGO [31,32].

Figure 2a,b show typical scanning electron microscope (SEM) images of the obtained ZnO and ZnO/rGO hybrid. In Figure 2a, ZnO NPs are aggregated to a certain extent. While in Figure 2b, rGO nanosheets serve as the support frame, ZnO NPs attach onto it densely and uniformly. Figure 2c,e show the transmission electron microscopy (TEM) images of our samples. As shown in Figure 2c, the particle size of ZnO is about 9.5 nm, while in Figure 2e, ZnO NPs are well dispersed on graphene sheets with a narrow particle size distribution of 6.5~8.5 nm. Note that ZnO NPs in ZnO/rGO are smaller than that of pure ZnO, due to the dispersing effect of rGO in preventing ZnO NPs from aggregation [33]. Figure 2d,f show the high resolution TEM (HRTEM) images of ZnO and ZnO/rGO, a clear lattice spacing of 0.28 nm, 0.25 nm and 0.19 nm can be observed, agrees well with the (100), (101) and (102) plane of hexagonal ZnO, respectively [34].

### 2.2. Electrocatalytic Activity towards Oxygen Reduction Reaction

The CV test was performed to evaluate the electro-activity of our samples in Ar and then in O_2_-saturated 0.1 M KOH solution from 0.2 to 0.8 V at a scan rate of 5 mV/s vs. Hg/Hg_2_Cl_2_ reference electrode. As shown in Figure 3a–c, the CV tests of ZnO, rGO and ZnO/rGO are essentially featureless under an Ar atmosphere. There is no peak appearing at the rGO electrode in O_2_-saturated electrolyte, either, indicating that rGO is inherently electrochemically silent under these conditions. A significant ORR peak is observed in ZnO and ZnO/rGO under an O_2_ atmosphere. Remarkably, ZnO/rGO exhibits a higher cathodic current density and a more positive ORR onset potential (−0.2 V vs. −0.24 V for ZnO) compared with ZnO (Figure 3d). The improved catalytic activity of ZnO/rGO can be attributed to the presence of rGO nanosheets, which improve the electrical conductivity of ZnO/rGO by providing a rapid electron transfer access [18]. The Tafel test results of ZnO and ZnO/rGO are illustrated in Figure 3e. The Tafel slopes are calculated to be 189 mV/dec and 133 mV/dec for ZnO and ZnO/rGO, respectively. It is obvious that the Tafel slope of ZnO/rGO is smaller than that of ZnO, the lower Tafel slope indicates the higher intrinsic catalytic activity [35].

The exchange current density (*i_0_*) on the two electrodes for ORR is calculated by Equation (1) (Appendix A) [36]. As shown in Table 1, the *i_0_* of ZnO and ZnO/rGO is calculated to be 6.76 × 10^−7^ and 9.21 × 10^−5^ mA/cm^2^, respectively. Obviously, the *i_0_* of ZnO/rGO is higher than that of ZnO, which is approximate to the commercial Pt/C (10^−6^~10^−9^ mA/cm^2^) [37,38]. The efficient electron transfer between ZnO NPs and rGO nanosheets would result in a high exchange current density and positive charge of ZnO/rGO [39]. To further highlight the benefits of rGO, EIS tests of ZnO and ZnO/rGO were performed at a potential of −0.30 V (vs. SCE). As shown in Figure 3f, the semicircle diameter of ZnO/rGO at the high frequency is smaller than that of ZnO, and the charge transfer resistance (R_ct_) can be evaluated accordingly. Apparently, ZnO/rGO reveals the smaller R_ct_, succeeded by ZnO. It is evident that rGO nanosheets can facilitate the process of ORR.

Rotating disc electrode (RDE) measurements of ZnO and ZnO/rGO were conducted to evaluate the electrocatalytic activity. The test of commercial Pt/C is also provided for comparison. As seen in Figure 4a,c,e, at the mixed kinetics-diffusion control region, the three catalysts show a well-defined increased cathodic current density, while in the diffusion-limited region they all show a parallel plateau.

Koutecky-Levich theory was employed to unravel the ORR mechanism of ZnO, ZnO/rGO and commercial Pt/C catalyst. Accordingly, the transferred electron numbers (n) can be calculated by Equations (2) and (3) (Appendix A) [40]. As shown in Figure 4b, the n of ZnO is calculated to be about 3.28 at a potential range of −0.5~−0.8 V, while in Figure 4d,f, the calculated n of ZnO/rGO and Pt/C are about 3.93 and 3.90, respectively. This result indicates that the ORR on ZnO/rGO and Pt/C electrode likely follows a 4-electron mechanistic pathway. However, the ORR catalyzed by ZnO runs in a mixed 2- and 4-electron mechanistic fashion [41]. Rotating ring disc electrode (RRDE) measurements are conducted to probe the exact ORR mechanism of the three catalysts. Figure 4g,i,k show the ring and disk currents of ZnO, ZnO/rGO and Pt/C, recorded at 1600 rpm at a scan rate of 5 mV/s in 0.1 M KOH solution. Compared with ZnO, ZnO/rGO exhibits a higher disk current, while a less amount of intermediate by-products during ORR process. But the disk current of ZnO/rGO is still smaller than that of Pt/C catalyst. Furthermore, the ORR percentage of peroxide species and electron transfer numbers of the three samples were calculated from Equations (4) and (5) (Appendix A) [42]. Figure 4h,j,l show the obtained results. In Figure 4h, the measured H_2_O_2_ yield on ZnO electrode is lower than 13%. Specifically, the average electron transfer numbers is found to vary between 3.1 and 3.4, indicating that the ORR of ZnO electrode proceeds via both 2- and 4-electron transfer pathways, while, the H_2_O_2_ yield on ZnO/rGO electrode is lower than 3%, accordingly, the average electron transfer numbers is found to vary between 3.9 and 4.0, indicating a 4-electron process of ORR. The H_2_O_2_ yield on Pt/C electrode is lower than 6%, and the average electron transfer numbers is found to vary between 3.5 and 4.1. These results are clearly shown in Figure 5a, suggest that the ORR catalyzed by ZnO/rGO is mainly follows a 4-electron process by directly forming hydroxyl species as the final products, which is similar to that of commercial Pt/C catalyst.

Chronoamperometry studies are widely used as an effective method to test catalytic stability [43]. Figure 5b shows the chronoamperometric measurement results of ZnO, ZnO/rGO and commercial Pt/C catalyst, by running the ORR for 16,000 s at −0.30 V. It is clearly seen that ZnO/rGO exhibits the highest relative current and a very slow attenuation rate, while commercial Pt/C shows a gradual degradation. For ZnO/rGO hybrid, the loss of catalytic activity results either from the structural changes of the rGO nanosheets or from the degradation of rGO during CV scans. The result clearly suggests the best durability of ZnO/rGO compared with ZnO and commercial Pt/C catalyst.

## 3. Materials and Methods

### Sample Preparation

Graphene oxide (GO) was prepared by a modified Hummers method in our laboratory [44,45], as illustrated in Scheme 1. The ZnO/rGO hybrid was synthesized by a facile hydrothermal method. GO (6 mL, 6 g/L) was added dropwise into the mixture of sodium hydroxide and zinc nitrate hexahydrate (Zn(NO_3_)_2_·6H_2_O). After magnetic stirring for 10 min, absolute ethyl alcohol and ethylene glycol were added into this mixture. The obtained solution was placed in a Teflon liner that was sealed in a stainless cylinder and kept at 180 °C for 24 h. After naturally cooling down, the precipitates formed were collected, washed with ethanol and deionized water. Finally, the obtained powder was fully dried at 60–80 °C. Thus 0.80 g of ZnO/rGO was obtained, and the yield of reaction was calculated to be about 94%. Moreover, ZnO catalyst and rGO catalyst were also prepared for comparison, under the same conditions.

## 4. Conclusions

In summary, we have synthesized a ZnO/rGO hybrid through a facile hydrothermal process. The obtained ZnO/rGO catalyst exhibits high cathodic current density and positive onset potential for ORR in alkaline media, due to the enhanced electron transfer kinetics resulting from the synergetic effect between ZnO NPs and rGO nanosheets. The electron transfer numbers at ZnO/rGO electrode during the ORR process is calculated to be 3.93, indicating a 4-electron mechanistic pathway from O_2_ to H_2_O. Besides, compared to commercial Pt/C catalyst, ZnO/rGO electrode also shows superior fuel durability, therefore, the ZnO/rGO hybrid is expected to have great potential as an effective noble-metal-free and low-cost catalyst for ORR.

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
