# Peer review of "Synthesis and Characterizations of Zinc Oxide on Reduced Graphene Oxide for High Performance Electrocatalytic Reduction of Oxygen"

_molecules, 2018, doi:10.3390/molecules23123227_

Reviewer 1 Report

This manuscript describes the preparation of zinc oxide modified reduced graphene via hydrothermal method for oxygen reduction reaction. Although authors conducted several characterizations and electrochemical analyses, some parts of manuscript deserve careful modifications.

           -Authors mentioned that the existence of inconspicuous peak at around 24°-In figure 4 on page 7, RDE and/or RRDE profiles of Pt/C are missing. Authors should supply these graphs as a reference.

          In figure 5 on page 8, the electron transfer numbers of Pt/C are also required.

Author Response

Dear reviewer,

 Thank you for your help during the review process. We have carefully read through the comments from you and would like to thank you for your very thoughtful comments and suggestions. We have accordingly revised the manuscript (highlighted in RED) to address your comments. The file below is our responses to your comments.

 We hope that the revised manuscript is now acceptable to Molecules. And we are looking forward to hearing from you soon.

Yours sincere,

Min Sun

Reviewer 2 Report

1) English should be refined and polished throughout the manuscript

2) Abstract should be clearly written in a different way

3) Introduction : I request the authors to write separate paragraph clearly for the proposed research activity. Authors should include as many as good references for graphene based metal oxide toward ORR. References need to be included extensively.

4) Sample preparation needs to be properly numbered

5) All the Figures need more clarity Fig 1 to Fig 5

6) Change full XPS to Survey spectra

7) Change SEM/TEM tests to SEM/TEM images of

8) Change attribute to attributed 

9) Tafel plot needs to replotted. The values reported are too high for the studied materials. Please refer relevant articles.

10) Suggestion : Revisit all figure captions

11) Where are the Equations ?

12) The whole R&D section needs to be revisited.

Author Response

Dear reviewer,

Thank you for your help during the review process. We have carefully read through the comments from you and would like to thank you for your very thoughtful comments and suggestions. We have accordingly revised the manuscript (highlighted in RED) to address your comments. The file below is our responses to your comments.

 We hope that the revised manuscript is now acceptable to Molecules. And we are looking forward to hearing from you soon.

 Yours sincere,

Min Sun

Round  2

Reviewer 1 Report

The authors have addressed all the concerns raised by the reviewer and it can be accepted to publish as it is.   

Author Response

Reviewer: The authors have addressed all the concerns raised by the reviewer and it can be accepted to publish as it is.  

Authors’ Response:

Dear reviewer:

Thank you for your kindness during the review process.

Wish you a good day.

Yours sincere

Min Sun

Reviewer 2 Report

Abstract : Correct last but one sentence. 

I do see some minor grammatical mistakes throughout the manuscript. For example, change obviously to obvious in page 5/11 (line number : 142). Also find similar occurrences and rectify the same. 

Author Response

Dear reviewer, Thank you for your kindness and positive suggestions during the review process. We have read through your suggestions and would like to thank you for your very careful examinations. We have accordingly revised the manuscript (this time, highlighted in Green) to address your comments. Below are our responses (marked as Authors’ Response following each comment from you) to your comments.

Reviewer: 1. Abstract: Correct last but one sentence.

Response: Thank you for your excellent suggestions. The last sentence of our Abstract has been rewritten. Page 1 line 24: The sentence of “It is suggested that ZnO/rGO hybrid is an efficient non-precious metal catalyst for oxygen reaction in alkaline medium” has been rewritten as “It is expected that the ZnO/rGO hybrid could be used as promising non-precious metal cathode in alkaline fuel cells”

Reviewer: 2. I do see some minor grammatical mistakes throughout the manuscript. For example, change obviously to obvious in page 5/11 (line number: 142). Also find similar occurrences and rectify the same.

Response: Thank you for your careful examination and excellent suggestions. We have checked the manuscript thoroughly and revised the grammatical mistakes.

Page 3 line 91: No evidence of any impurities were detected. (change “is found” to “were detected”) Page 3 line 100: The sentence of “On the surface of ZnO and ZnO/rGO, the element of C, O and Zn is detected” has been rewritten as “As expected, Fig. 1c clearly showed that the ZnO/rGO mainly contains the C, O and Zn elements”

Page 4 line 103: Fig. 1d shows the C 1s spectrum of ZnO/rGO hybrid,…. (change “spectra” to “spectrum”)

Page 4 line 121: In Fig. 2a, ZnO NPs are aggregated to a certain extent. (change “is” to “are”)

Page 5 line 142: It is obvious that the Tafel slope of ZnO/rGO is smaller than that of ZnO,… (change “obviously” to “obvious”)

Page 8 line 174: …the calculated n of ZnO/rGO and Pt/C are about 3.93 and 3.90, …(change “is” to “are”)